# Extracellular Vesicles as Surrogates for the Regulation of the Drug Transporters ABCC2 (MRP2) and ABCG2 (BCRP)

**DOI:** 10.3390/ijms25074118

**Published:** 2024-04-08

**Authors:** Juan Pablo Rigalli, Anna Gagliardi, Klara Diester, Gzona Bajraktari-Sylejmani, Antje Blank, Jürgen Burhenne, Alexander Lenard, Lars Werntz, Andrea Huppertz, Lena Münch, Janica Margrit Wendt, Max Sauter, Walter Emil Haefeli, Johanna Weiss

**Affiliations:** 1Department of Clinical Pharmacology and Pharmacoepidemiology, Medical Faculty Heidelberg, Heidelberg University Hospital, Heidelberg University, Im Neuenheimer Feld 410, 69120 Heidelberg, Germanywalter-emil.haefeli@med.uni-heidelberg.de (W.E.H.); johanna.weiss@med.uni-heidelberg.de (J.W.); 2Department of Food and Drug, University of Parma, Parco Area delle Scienze 27/A, 43124 Parma, Italy; 3MVZ Diaverum Remscheid, Rosenhügelstraße 4a, 42859 Remscheid, Germany

**Keywords:** ABC transporters, ABCC2, ABCG2, pharmacokinetics, multidrug resistance, chemoresistance, exosomes, extracellular vesicles

## Abstract

Drug efflux transporters of the ATP-binding-cassette superfamily play a major role in the availability and concentration of drugs at their site of action. ABCC2 (MRP2) and ABCG2 (BCRP) are among the most important drug transporters that determine the pharmacokinetics of many drugs and whose overexpression is associated with cancer chemoresistance. ABCC2 and ABCG2 expression is frequently altered during treatment, thus influencing efficacy and toxicity. Currently, there are no routine approaches available to closely monitor transporter expression. Here, we developed and validated a UPLC-MS/MS method to quantify ABCC2 and ABCG2 in extracellular vesicles (EVs) from cell culture and plasma. In this way, an association between ABCC2 protein levels and transporter activity in HepG2 cells treated with rifampicin and hypericin and their derived EVs was observed. Although ABCG2 was detected in MCF7 cell-derived EVs, the transporter levels in the vesicles did not reflect the expression in the cells. An analysis of plasma EVs from healthy volunteers confirmed, for the first time at the protein level, the presence of both transporters in more than half of the samples. Our findings support the potential of analyzing ABC transporters, and especially ABCC2, in EVs to estimate the transporter expression in HepG2 cells.

## 1. Introduction

Drug transporters of the ABC (ATP-binding cassette (ABC)) superfamily mediate the efflux of a variety of drugs and, therefore, have a major impact on drug distribution, clearance, and systemic bioavailability. In general, ABC transporters are physiologically expressed in epithelia of pharmaco-toxicological relevance, such as the brush-border membrane of the intestine, the bile canaliculi, the apical membrane of the renal proximal tubules, and the blood–brain barrier [1]. Due to their frequent overexpression in tumor cells, ABC transporters have been associated with an increased efflux and reduced intracellular concentration of anticancer drugs and, ultimately, treatment failure [1,2]. Together with P-glycoprotein (P-gp, ABCB1, MDR1), the later-discovered multidrug-resistance associated protein 2 (MRP2, ABCC2) and breast-cancer-resistance protein (BCRP, ABCG2) are among the most relevant ABC transporters involved in drug efflux [1,3].

ABCC2 has been reported to transport penicillin, saquinavir, indinavir, ritonavir [4], cisplatin, anthracyclines, vinca alkaloids, methotrexate, epipodophyllotoxins, and sorafenib, among other drugs [5,6]. Therefore, changes in ABCC2 expression and activity can affect drug exposure and have an impact on the efficacy and toxicity of the treatment. For example, ABCC2 up-regulation was associated with a decrease in the plasma area under the curve (AUC) of its substrate ezetimibe and related glucuronide metabolite [7]. Similarly, the up-regulation of ABCC2 resulted in a lower plasma AUC of its substrates: mycophenolic acid [8] and carvedilol [9]. In addition, ABCC2 has been detected in various solid tumors, such as hepatocellular carcinoma (HCC) [10], head and neck squamous cell carcinoma (HNSCC) [11], esophageal squamous cell carcinoma (ESCC) [12], papillary renal cell carcinoma [13], and colon cancer [14]. An inverse association between ABCC2 expression in the tumor and chemotherapy response and/or disease prognosis has been established in clinical trials. This was the case, for example, in HCC patients treated with cisplatin [15] and ESCC patients treated with cisplatin, doxorubicin (DOX), and 5-fluorouracil [12]. Here, however, it should be noted that a causal relation between the sole transporter overexpression and therapy resistance has not been established and other proteins may also contribute to the described chemoresistance.

ABCG2 transports several drugs, including antibiotics (e.g., ciprofloxacin, enrofloxacin, norfloxacin), antiviral agents (e.g., zidovudine, abacavir) [1], statins (e.g., rosuvastatin, simvastatin, atorvastatin) [16,17], sulfasalazine, and direct oral anticoagulants (e.g., apixaban, rivaroxaban), with the latter exhibiting a particularly narrow therapeutic index. Chemotherapeutic agents, including DOX [18], 5-fluorouracil [19], erlotinib, etoposide, gefitinib, methotrexate, mitoxantrone, and imatinib [1], are also ABCG2 substrates. ABCG2 is physiologically expressed in the liver, small intestine, kidney, blood–brain barrier, and placenta [1]. To date, the potential role of ABCG2 in the pharmacokinetics of its substrates has mainly been established by analyzing the effect of different single-nucleotide polymorphisms [17]. In addition, the co-administration of ABCG2 inhibitors such as ketoconazole and ritonavir increased the exposure to the ABCG2 substrate rivaroxaban [20]. In addition to its role under physiological conditions, ABCG2 has been associated with the chemoresistance of several cancers [11,21,22,23]. A discussion on the relative contribution of ABC transporters to drug resistance and their interpretation can be found in the article by Piet Borst [24]. Overall, ABCG2 and ABCC2 can reduce tissue exposure to drugs, not only by leading to lower (intracellular) drug concentrations at the site of action but also by reducing the amount of drug reaching the tissues. However, the findings are limited due to the lack of tissue expression data, which is mainly because biopsy sampling is very invasive.

The expression and, therefore, the activity of ABCC2 and ABCG2 are regulated by drugs, environmental toxicants, hormones, diet [25,26], and viral proteins [27,28], among other factors. In this way, changes in the bioavailability and accumulation of transporter substrates can be expected during treatment. In general, the molecular mechanisms underlying transporter regulation are complex and involve the participation of nuclear receptors, coactivators, and corepressors, with the arrangement of these being highly variable between individuals [26,29]. As an example, more than 100-fold interindividual variability has been reported in hepatic ABCC2 [30]. Interindividual differences have also been reported in ABCG2 protein expression [31]. This interindividual variability in healthy and tumor tissues, as well as the variable and sometimes unpredictable extent of transporter regulation before and during treatment, represents an important challenge that must be overcome to ensure the efficacy and safety of therapies. To date, repeated biopsy sampling is the only way to monitor ABC transporter expression in the relevant tissues. However, due to its invasive nature, this is not performed routinely or in cases where serious side effects would be expected as a result of inappropriate dosing. In addition, adjuvant chemotherapy in cancer patients usually targets residual tumor cells and/or distant metastases where a biopsy is not technically possible.

Extracellular vesicles (EVs) are nanoparticles released by all cell types of the organism. They mostly comprise exosomes and microvesicles [32]. Under physiological conditions, EVs mediate cell–cell communication [32]. The cargo of EVs consists mostly of proteins, RNA, and lipids, and the arrangement of these components often results in a close fingerprint to the cell of origin [32]. Furthermore, changes in the cells of origin can lead to changes in the composition of the EVs they release [33,34], thus making EVs a promising source of biomarkers. The presence of EVs from a wide range of tissues in blood, as well as renal EVs in urine, allows for simple access to these nanoparticles in clinical practice. Therefore, valuable tissue information could be obtained in a minimally invasive way. While different studies already confirmed the presence of drug-metabolizing enzymes [35] and P-gp [36], so far, ABCC2 and ABCG2 have only been determined in plasma EVs at the mRNA level [37], which does not necessarily correlate with the transporter activity. It is still unknown whether the amount of ABCC2 and ABCG2 in EVs, in general, can be used to specifically predict changes in the transporter expression and activity in the cells of origin upon exposure to modulators, as would therapy itself, co-administered drugs, or other regulatory factors. Hence, the aim of this project was to assess the potential of EVs as dynamic surrogates for ABCC2 and ABCG2 expression and activity in vitro and in healthy volunteers.

## 2. Results

### 2.1. UPLC-MS/MS Method Validation

Surrogate peptides were well separated by the established chromatographic method with retention times of 4.3–4.4 min for SSL and 4.8–4.9 min for LTI, showing peak widths of 4 s. Representative chromatograms are shown in Appendix A for LTI and Appendix A for SSL. Inter- and intraday accuracy and precision are presented in Table 1. Recovery was within 90.4 and 110.7%, and therefore consistent within QC levels, which demonstrates the suitability and efficiency of the established extraction procedure. IS-normalized matrix effects between 85.3 and 100.1% confirmed the feasibility of using the isotopologues of the surrogate peptides IS. Correlation coefficients (R^2^) for the linear regression within the calibration range (25–100,000 pg/mL) were >0.99 for both surrogate peptides in all validation batches. Furthermore, for SSL, a calibration sample (12.5 pg/mL) below the original LLOQ was measured using six-fold determination in two independent batches to demonstrate the possibility of using this limit for quantification. The accuracy was >88.6%, and therefore in line with the performance achieved for the more concentrated samples. The coefficient of variation (i.e., precision) was <21.3%, only slightly above the target value of ≤20%.

### 2.2. EV Characterization

The particle size distribution of all EV samples from cell cultures was determined for all the treatments by dynamic light scattering (DLS). The results (mean < 350 nm) indicated enrichment with small EVs and were compatible with the specifications of the size exclusion chromatography columns (Appendix A). Plasma EVs from both studies also showed a size distribution compatible with small EVs. Namely, the size of EVs from the Phimax study was 119 ± 26 nm before and 131 ± 29 nm during treatment with St. John’s wort (SJW) (*n* = 13). The size of EVs from the ApeX study was 95 ± 12 nm (*n* = 11) before and 83 ± 18 nm (*n* = 10) during treatment with carbamazepine (CMZ). In addition, the presence of the EV marker CD63 was confirmed by Western blot in EVs derived from HepG2, MCF7 cells, and plasma (Appendix A). CD81 was detected in MCF7-derived EVs, and a slight signal was observed in plasma EVs. No CD81 signal was detected in HepG2 EVs (Appendix A). In fact, frequently, different types of cells may release EVs carrying different surface markers [38]. Therefore, more than one marker is usually analyzed, as performed in our study. The detection of at least one surface marker in each type of preparation, plus the confirmation of the expected particle size, as determined by DLS, confirm the success of our EV isolation protocol.

### 2.3. EVs as Surrogate for ABCC2 and ABCG2 Regulation in Cell Lines

HepG2 cells were used as a model, with high ABCC2 expression (Appendix A). From all the investigated treatments, only rifampicin and hypericin exhibited an effect on ABCC2 protein levels. No changes in ABCC2 protein expression were observed in HepG2 cells treated with hyperforin, enzalutamide, or mitotane (Appendix A). Treatment with rifampicin (5–20 μM) resulted in ABCC2 down-regulation at the protein level at 48 h (Figure 1a) and 72 h of treatment (Figure 1b), thus confirming previous reports on the same cell line [39]. In line with the decreased protein expression, rifampicin also decreased ABCC2 activity at 48 h (Figure 1c) and 72 h of treatment (Figure 1d), as determined by measuring the efflux of DCF. An analysis of ABCC2 in EVs revealed no changes in the transporter expression after 48 h of exposure to rifampicin (Figure 1e). At 72 h of treatment, a decrease in ABCC2 expression in the EVs of cells treated with rifampicin 10 and 20 μM was observed (Figure 1f). In addition, the treatment of HepG2 cells with hypericin (0.01–0.5 μM, 72 h) resulted in the down-regulation of ABCC2 protein expression (Figure 2a) and activity (Figure 2b). EVs from treated cells did reproduce the decrease in ABCC2 cellular protein levels via hypericin (Figure 2c). Altogether, our data indicate a clear association of ABCC2 expression and activity levels in the cells and the amount of the transporter in the derived EVs.

MCF7 cells were used as the model with high ABCG2 expression in cells (Appendix A) and EVs (Appendix A). MCF7 cells treated with rifampicin (20 μM, 48 h) exhibited significant ABCG2 up-regulation at the protein level (Figure 3a). A similar effect was observed at 72 h of treatment, although it only reached statistical significance for 5 and 10 μM rifampicin (Figure 3b). ABCG2 activity did not show any significant changes at 48 h of treatment (Figure 3c). At 72 h of exposure, only treatment with rifampicin 10 μM led to an increase in the transporter activity (Figure 3d). MCF7 EVs did not show any changes in ABCG2 levels at 48 h (Figure 3e) or 72 h of treatment with rifampicin (Figure 3f). Since hyperforin, hypericin, enzalutamide, and mitotane did not alter ABCG2 protein expression in MCF7 cells (Appendix A), no comparisons between cells and EVs were made for these compounds.

To rule out the influence of cell death or decreased growth on the observed effects, cell viability was quantified for those treatments for which changes in ABCC2 or ABCG2 were observed by staining with crystal violet. Neither rifampicin nor hypericin affected the viability of HepG2 or MCF7 cells (Appendix A).

### 2.4. ABCC2 and ABCG2 Analysis in Plasma EVs

EVs isolated from plasma samples obtained in the ApeX study exhibited detectable ABCC2 expression in 6 out of 12 study participants before treatment with CMZ and in 2 out of 12 participants during treatment. Mean ABCC2 expression levels in EV samples were 0.472 ± 0.328 and 0.170 ± 0.141 pg LTI/μg EV protein before and during CMZ, respectively (Figure 4a). Expression levels varied greatly between participants (CV: 69.6% before and 83.2% during treatment). In the same study, ABCG2 was detected in plasma EVs of 10 of the 12 participants before treatment and 8 of the 12 participants during CMZ. The expression levels were 0.113 ± 0.086 before and 0.091 ± 0.099 pg SSL/μg EV protein during CMZ (Figure 4c). The CV was 75.9% before and 108.4% during CMZ. It was not possible to conduct a paired analysis for ABCC2 since only one participant exhibited detectable transporter expression before and during treatment (Figure 4b). A paired analysis for ABCG2 did not show significant differences before or during treatment (Figure 4d).

From a total of 13 participants of the Phimax trial, ABCC2 was detected in 9 participants (0.810 ± 1.105 pg LTI/μg EV protein, CV: 136.4%) before initiation of the treatment with SJW, and in 10 participants at the time of completion of the treatment (0.767 ± 0.858 pg LTI/μg EV protein, CV: 111.9%) (Figure 4e). ABCG2 was detected in 10 participants before the treatment (0.078 ± 0.106 pg SSL/μg EV protein, CV: 136.9%) and in 11 participants during the treatment (0.011 ± 0.008 pg SSL/μg EV protein, CV: 77.82%) (Figure 4g). A paired analysis of ABCC2 (Figure 4f) and ABCG2 expression (Figure 4h) before and during treatment did not show any significant differences due to the treatment with SJW. Altogether, our findings confirm the presence of ABCC2 and ABCG2 at the protein level in the plasma EVs of a group of study participants. However, due to the high variability, no significant differences before and after any of the treatments were found.

## 3. Discussion

Drug transporters of the ABC family play a major role in drug absorption, distribution, and excretion in health and disease. They determine the systemic bioavailability and the intracellular concentration of drugs at the site of action. This is particularly the case for tumor cells, which frequently exhibit an overexpression and up-regulation of ABC transporters. Higher transporter expression and activity are frequently associated with increased efflux, and thus a lower accumulation, e.g., of chemotherapeutic agents. ABCC2 and ABCG2 are two members of the ABC transporter family that have been widely identified in tumor cells and have been associated with response to therapy and disease prognosis [10,11,12,13,14,23]. In addition, ABCC2 and ABCG2 are expressed very differently between individuals [30,31] and are regulated by co-administered drugs, hormones, and diet [26]. Therefore, monitoring changes in the transporter expression in relevant tissues, especially in tumor cells, could help to avoid treatment failure or toxicity. Simple and non-invasive alternatives to repeated biopsies are needed for this purpose. Due to their presence in most biological fluids [40] and their protein content, reflecting changes in the proteome of the cell of origin [33,34], EVs seem to be a promising tool to assess changes in ABCC2 and ABCG2 expression. However, the quantification of ultra-low protein levels in EVs requires highly sensitive technologies. Here, we developed and validated a UPLC-MS/MS method for the simultaneous quantification of ABCC2 and ABCG2 at the protein level. Our method exhibited linearity over a large concentration range (Section 2.1), which allowed for its application to different types of biological samples with highly different expression levels, such as two different cell lines: EVs from cell culture and EVs from plasma. In addition, a previous study demonstrated the better performance of UPLC-MS/MS compared to Western blot in the detection of ABCG2 [41]. Furthermore, in the case of EV samples, Western blot analysis lacks appropriate housekeeping proteins and, therefore, is unlikely to deliver reliable quantitative data. Considering that both transporters can be regulated simultaneously and in a coordinated manner [42], their concomitant quantification in a single preparative and analytical approach (i.e., in a single sample), as is the case for UPLC-MS/MS, can help to reduce the amount of material required, as well as the associated costs and processing times.

In order to investigate whether EVs reflect changes in the transporter expression and activity of the cells of origin, situations of transporter regulation in our experimental models had to be identified. We observed a down-regulation of ABCC2 via rifampicin (Figure 1) and hypericin (Figure 2) in HepG2 cells, which is in line with previous reports [39,43,44,45]. Under our experimental conditions and in accordance with previous studies, the translational [39] and post-translational down-regulation of ABCC2 [44] via rifampicin clearly outweighed the well-known pregnane X receptor-dependent transcriptional induction [46]. We also observed a decrease in ABCC2 activity, in line with the lower protein expression of this transporter. Notably, DCF may also be transported by other proteins (e.g., ABCC3). However, previous findings suggest a major role of apical efflux (i.e., via ABCC2) with respect to basolateral transport [47]. A UPLC-MS/MS analysis of EVs derived from HepG2 cells revealed a significant down-regulation of ABCC2 only at 72 h of treatment. In the case of rifampicin, for which transporter regulation in the cells was observed at an earlier time-point, this clearly points to the later onset of the effects in EVs compared to the cells of origin. A previous study reported a half-life of the ABCC2 protein in rat liver of 22–36 h [48] and EV biogenesis was demonstrated to require at least 8–16 h [49,50]. Once released, EV stability after 24 h at 37 °C may reach more than 90% [51]. Hence, the delay in the decrease in ABCC2 protein levels in EVs compared to the cells of origin is likely to be the consequence of the long ABCC2 half-life, EV biogenesis, and EV stability time. At 48 h of treatment, the EV population consists mostly of particles released hours or even a day before the EV collection, a time frame at which it is most likely that no down-regulation of ABCC2 has occurred. Moreover, once the down-regulation has occurred in the cells of origin, the population of EVs released earlier will still mask the decrease in ABCC2 concentration that occurred in the meantime due to the long half-life of the EVs and the ABCC2 present in their cargo. Notably, in addition to ABCC2, the half-life of membrane drug transporters in vitro and in vivo may be longer than 12 h, as demonstrated for P-gp [52], and could even reach more than a day, as for ABCG2 [53]. Although the transporter half-life and EV stability should be taken into consideration to detect transporter down-regulation during treatment, our findings clearly highlight the potential of EVs as surrogates for ABCC2 transporter levels in HepG2 cells. Studies in more complex models (e.g., primary hepatocytes) should be performed to elucidate the translational potential of these findings.

In addition, our data show ABCG2 up-regulation via rifampicin in MCF7 cells, which partly translates into the higher efflux of DOX (Figure 3), a cytostatic agent partially transported by ABCG2. These findings are also in line with a previously demonstrated up-regulation of ABCG2 via rifampicin in primary hepatocytes [42]. However, contrary to the findings for ABCC2 in HepG2 cells, EVs do not reflect this ABCG2 up-regulation at either time point. Based on evidence from other models, significant EV biogenesis and release are expected to take place within 24 h [49]. Hence, the cargo of the EVs at the 72 h time point, at the latest, is expected to represent the situation in the cells at 48 h, when a significant ABCG2 up-regulation was already established (Figure 3). Interestingly, in the pancreatic cancer cells PANC-1, a clear dissociation between ABCG2 protein levels in EVs and cell lysates was reported [54]. Similarly, in MDA-MB-231 breast cancer cells, increasing levels of ABCG2 in EVs, concomitant with decreasing levels of this protein in the cells of origin, were observed [55]. In fact, the sorting of proteins into the EVs is a highly complex and regulated process [56], in which the cargo of the vesicles may not always reflect the changes in the cells of origin. In addition, a different composition of the EVs released by HepG2 and MCF7 cells, as evidenced by the differences in the expression of the EV marker CD81 (Appendix A), is possible. Differences in the sorting of tetraspanins, such as CD63 and CD81, have already been reported in other models [38]. Thus, our findings suggest the lower potential of EVs to estimate changes in ABCG2 levels in the cells of origin, at least via rifampicin. Moreover, our results highlight the relevance of performing individual assessments for each protein of interest in EVs and the cells of origin.

To further investigate the potential of EVs as biomarkers for drug efflux in vivo, we determined the levels of ABCC2 and ABCG2 in total plasma EVs from healthy volunteers before and during treatment with the known inducers CMZ or SJW. In previous clinical trials, CMZ has been reported to induce duodenal ABCC2 at the protein level [57]. Additionally, a large body of evidence, mostly in different in vitro and animal models, supports the activation of the pregnane X receptor via the main active principle of SJW (i.e., hyperforin) [58] and, in this way, the regulation of ABCC2 [59] and ABCG2 [60]. In a randomized clinical trial, duodenal up-regulation of the closely related transporter P-gp via SJW was observed [61]. In our study, with the exception of ABCC2 in the ApeX cohort, both transporters were detected in plasma EVs of more than half of the study participants. However, due to the high interindividual variability, no significant changes before and after treatment were observed in EVs, failing to reflect effects previously demonstrated in the tissues of origin [57,58,59]. This could be attributed, at least partially, to the extremely low proportion of EVs from the tissue(s) with transporter regulation compared to the total amount of circulating EVs. According to previous studies, more than 99% of plasma EVs derive from hematopoietic cells [62], some of which also express ABCC2 and ABCG2 [63] but may not be subject to regulation by SJW or CMZ. In this regard, the enrichment of tissue-specific samples using affinity methods [64,65] or the calculation and normalization to a tissue-shedding factor [37] may allow for the detection of effects that are otherwise undetectable in whole plasma EVs.

The evidence obtained to date demonstrated the presence of ABCC2 and ABCG2 at the protein level in EVs in vitro, but did not address whether these protein levels vary in response to changes in the cells of origin. This is a major aspect of assessments of the biomarker potential of EVs. Furthermore, in plasma EVs, ABCC2 and ABCG2 were only detected at the mRNA level [37]. In the first part of this study, we clearly demonstrated the association between ABCC2 protein levels in EVs and the protein expression and activity in HepG2 cells. Protein analyses of EVs may benefit from their higher molecule stability compared to mRNA molecules. Furthermore, since activity measurements in the EVs may be biased by the occasional reverse topology of membrane transporters in EVs [38], the determination of the transporter levels in EVs appears to be the closest surrogate available to the transporter activity in the cells of origin. In this regard, we detected the presence of ABCC2 and ABCG2, also at the protein level, in plasma EVs from two cohorts of healthy volunteers, representing the first study to quantify both proteins in EVs in vivo. Although our study provides the first piece of evidence associating ABCC2 protein levels in cells and EVs, further studies should be performed to elucidate whether this association also takes place in other cell models and in vivo and, therefore, assess the clinical relevance of our findings.

## 4. Materials and Methods

### 4.1. Materials

DMEM and RPMI1640 media were from Pan-Biotech (Aidenbach, Germany). Fetal calf serum (FCS) was from Capricorn Scientific (Ebsdorfergrund, Germany). Aprotinin, 5(6)-carboxy-2′,7′-dichlorofluorescein diacetate, MK571, fumitremorgin C (FTC), Hank’s buffered saline solution (HBSS), hypericin, phosphate buffered saline (PBS), trypsin, penicillin/streptomycin for cell culture, and L-glutamine were from Sigma-Aldrich (Taufkirchen, Germany). Hyperforin, hypericin, mitotane, and enzalutamide were from Santa Cruz Biotechnology (Heidelberg, Germany). Bovine serum albumin, dimethyl sulfoxide (DMSO), rifampicin, and Triton-X100 were from AppliChem GmbH (Darmstadt, Germany). Chloroform was from Merck (Darmstadt, Germany). Acetone was from Carl Roth GmbH (Karlsruhe, Germany). Pefabloc was from Serva (Heidelberg, Germany). Mini-Protean TGX Precast gels 12% were from Bio-Rad (Hercules, CA, USA). Acetonitrile, isopropanol, methanol, and formic acid for ultra-performance liquid chromatography (UPLC) were from Biosolve Chimie (Dieuze, France). Ultra-pure water for UPLC was prepared with an arium^®^ mini (Sartorius, Göttingen, Germany) ultrapure water system. Leupeptin and pepstatin A were from Biomol (Hamburg, Germany). RIPA buffer and the Pierce BCA assay kit were from ThermoScientific (Waltham, MA, USA). DOX was obtained from Biotrend Chemikalien GmbH (Köln, Germany). The ProSieve QuadColor protein marker used for SDS-PAGE (#00193837) was from Lonza (Köln, Germany). Low binding tubes (Biozym Scientific GmbH, Oldendorf, Germany) were used for all the steps involving EVs and proteins.

### 4.2. Cell Lines and Treatments

MCF7 cells (human-estrogen-sensitive breast cancer cells expressing ABCG2) were from the European Collection of Authenticated Cell Cultures (Salisbury, United Kingdom). HepG2 cells (human hepatocellular carcinoma cells expressing ABCC2) are available at the American Tissue Culture Collection (Manassas, VA, USA). MCF7 cells were cultured in DMEM supplemented with 10% FCS, 2 mM L-glutamine, 100 U/mL penicillin, 100 µg/mL streptomycin, and HEPES (10 mM). HepG2 cells were cultured in RPMI1640 medium supplemented with 10% FCS, 2 mM L-glutamine, 100 U/mL penicillin, and 100 µg/mL streptomycin. Cells were maintained at 37 °C with 5% CO_2_.

To investigate whether EVs reproduce changes in the transporter expression, the situations of transporter regulation in the cells of origin had to be characterized first. HepG2 (1.6 × 10^5^ cells/cm^2^) and MCF7 cells (5.7 × 10^4^ cells/cm^2^) were plated in 175 cm^2^ culture flasks (2.8 × 10^7^ cells/flask and 1 × 10^7^ cells/flask, respectively), cultured for 24 h, and treated with the well-known modulators rifampicin (5, 10 and 20 µM, 48 and 72 h) [39,42], hyperforin (0.01, 0.1 and 0.5 µM, 48 h) [43,60], hypericin (0.01, 0.1 and 0.5 µM, 72 h) [43], enzalutamide (0.8, 4 and 8 µM, 96 h) [66], and mitotane (10, 20 and 40 µM, 96 h) [67]. All compounds have been previously described to modulate ABCC2, ABCG2, or related transporters after a similar incubation time, albeit under different culture conditions (e.g., FCS-containing medium throughout the treatment) or in different cells [39,42,43,60,66,67]. To prevent sample contamination with EVs from FCS, 24 h prior to the end of the treatment, cells were washed with PBS, and an FCS-free treatment medium containing the corresponding concentration of the above-mentioned compounds was added to the cells. After treatment, cells were detached and lysates were prepared as described below. Transporter expression was analyzed by UPLC coupled with tandem mass spectrometry (UPLC-MS/MS). When changes in the transporter expression in the cells were observed, EVs were isolated from the culture medium, as described below.

### 4.3. Plasma Samples

Plasma samples were collected from healthy volunteers enrolled in two previous clinical trials at the Early Clinical Pharmacology Trial Unit (KliPS, Heidelberg University Hospital). These trials were the ApeX study (EudraCT 2018-002490-22, *n* = 12) [68] and the Phimax study (EudraCT 2016-002300-61, *n* = 13) [69]. In the ApeX study, plasma samples were obtained before and at the end of the treatment with carbamazepine (CMZ, target dose 200 mg/day, 21 days, p.o.). In the Phimax study, plasma samples were obtained before and on the last day of the treatment with St. John’s wort (SJW, 3 × 300 mg/day, 13 days, p.o.). Both CMZ and SJW are well-known transporter inductors [57,58,59,60]. Because the second sample was obtained at the end, but still within the duration of the treatment, the denominations “during SJW” and “during CMZ” were used throughout the manuscript and the figures. The original studies (Ref. AFmo-144/2019; Ref. AFmo-440/2016) and the use of the remaining samples for this study (Ref. S-147/2023) were approved by the competent authorities and had the positive vote of the Ethics Committee of the Medical Faculty of Heidelberg University. All studies were conducted according to the Declaration of Helsinki and the Good Clinical Practices. All study participants provided their written informed consent. Plasma samples were thawed and centrifuged at 2000× *g* (20 min, room temperature). Supernatants were further cleared by centrifugation at 10,000× *g* (20 min, room temperature). Supernatants from the last centrifugation step were divided into 500 µL aliquots and used for EV isolation (see Section 4.5).

### 4.4. Cell Lysates

After completion of the treatment, cells were detached by trypsinization, centrifuged (300× *g*, 5 min, room temperature), washed with PBS, and lysed by incubation with PBS supplemented with Triton-X100 (0.5% *v*/*v*) for 30 min. Lysates were clarified by centrifugation at 16,000× *g* (15 min, 4 °C). Supernatants were subjected to protein precipitation by incubation with acetone (60 min, −20 °C). Total cellular proteins were pelleted by centrifugation at 16,000× *g* (15 min, 4 °C), resolubilized in NaOH 1 M, and diluted 1:10 in Tris-EDTA buffer (50 mM/0.4 mM, pH 7.80). Protein concentration was determined by the BCA assay.

### 4.5. EV Isolation

After the last 24 h of treatment, culture medium containing the EVs was collected and centrifuged at 2000× *g* (30 min, 4 °C). Supernatants containing the EVs were concentrated using Amicon Ultra-15 PLHK membrane Ultracel-PL 100 kDa ultrafiltration devices (Merck, Darmstadt, Germany). In this way, 500 µL of a concentrate containing EVs was generated and loaded onto a qEV original 350 nm size exclusion chromatography column (IZON Science, Christchurch, New Zealand). Plasma samples were centrifuged as described in Section 4.3 and applied to the column. PBS, filtered through a 0.22 µm filter (Merck, Darmstadt, Germany), was used as the mobile phase. EVs were collected in the 2.4–3.6 mL fraction, according to the manufacturer’s specifications. The size distribution of the particles was determined by dynamic light scattering using a Zetasizer Nano ZS (Malvern Panalytical, Malvern, UK). EV suspensions were used for Western blotting and UPLC-MS/MS.

### 4.6. Western Blotting

In order to confirm the presence of EVs, the EV surface markers CD63 [70] and CD81 [71] were analyzed by Western blot. For this purpose, the EV-containing elution fraction (see Section 4.5) was concentrated using Amicon^®^ ultra filters 2 mL 3 kDa (Merck, Darmstadt, Germany). Concentrates were added with RIPA lysis buffer supplemented with pefabloc (1 mg/mL), aprotinin (1 µg/mL), pepstatin A (1 µg/mL), and leupeptin (5 µg/mL) as protease inhibitors. To compare the expression of ABCC2 and ABCG2 in different cell types, cellular proteins were purified as described in 4.4. Proteins were separated by SDS-PAGE and blotted onto a PVDF membrane. Subsequently, membranes were blocked with 5% bovine serum albumin for at least 1 h and incubated overnight with the primary antibodies anti-CD63 (sc-5275, 1:200 in blocking buffer), anti-ABCG2 (BXP21, 1:100 in blocking buffer) from Santa Cruz Biotechnology (Heidelberg, Germany), anti-CD81 (555675, 1:200 in blocking buffer) from BD Biosciences (Heidelberg, Germany), or anti-ABCC2 (M2-III-6, 1:100 in blocking buffer, Enzo Life Sciences, Farmingdale, USA). Mouse IgG HRP-Linked Whole Antibody (1:2000, Cytiva, Marlborough, MA, USA) was used as the secondary antibody. Immunoreactive bands were detected by chemiluminescence using the ChemoStar touch device (Intas Science Imaging Instruments GmbH, Göttingen, Germany).

### 4.7. EV Protein Purification

To analyze the presence and levels of ABC transporters in EVs, vesicle suspensions were lyophilized overnight and subjected to protein precipitation with methanol/chloroform (3:1 *v*/*v*). Protein pellets were resuspended in Tris-EDTA (50 mM/0.4 mM, pH 7.80) buffer, and protein concentration was determined by BCA assay.

### 4.8. Trypsin Digestion

Resuspended cellular proteins (see Section 4.4), EV proteins (see Section 4.7), and a surrogate matrix (bovine serum albumin, 0.4 mg/mL in Tris-EDTA) were digested using Trypsin Gold^®^ (Promega, Mannheim, Germany) at a trypsin to total protein ratio of 1:10 *w*/*w* (overnight, 37 °C).

### 4.9. ABC Transporter Analysis by UPLC-MS/MS

ABCC2 and ABCG2 were quantified in cells and EVs by UPLC-MS/MS using surrogate peptide quantification after trypsin digestion. Surrogate peptides were H-LTIIPQDPILFSGSLR-OH for ABCC2 (LTI, [72]) and H-SSLLDVLAAR-OH for ABCG2 (SSL, [41]). Purified peptides were acquired from Peptide Specialty Laboratories GmbH (Heidelberg, Germany).

Calibration samples (sample concentrations of 25, 50, 100, 300, 1000, 3000, 10,000, 30,000, and 100,000 pg/mL) and quality control samples (QCs) at sample concentrations of 25 (lower limit of quantification (LLOQ)), 75 (low QC), 37,500 (mid QC), and 75,000 pg/mL (high QC) were generated in a water–acetonitrile –formic acid ratio of 95:5:0.1% from independent weighings via the addition of 25 µL of spike solutions containing both peptides to the 100 µL surrogate matrix. Stable isotope-labelled peptides H-LTIIPQDPILFSGS[(^13^C_6_,^15^N)-L]R-OH and H-SS[(^13^C_6_,^15^N)-L]LDVLAAR-OH (Peptides & Elephants GmbH, Hennigsdorf, Germany) were used as internal standards (IS) for ABCC2 and ABCG2, respectively. Calibration and QC samples, as well as digested cellular and EV protein samples, were spiked with 25 µL IS solution. Peptides were purified from samples via solid-phase extraction using Oasis^®^ MAX µElution plates 30 µm (Waters, Milford, MA, USA). For this purpose, samples were alkalinized with an equal volume of aqueous NH_3_ (10% *v*/*v*) and loaded onto the extraction plate. Then, plates were washed with 100 µL of UPLC-quality water and, subsequently, 100 µL of methanol. Purified peptides were eluted with acetonitrile–methanol–water (2:1:1, *v*/*v*) supplemented with 5% (*v*/*v*) formic acid, with two additions of 30 µL. Extracts were diluted with 150 µL of UPLC-quality water.

The UPLC-MS/MS settings consisted of an Acquity Classic UPLC System equipped with an Acquity UPLC^®^ Peptide BEH C18 column (300 Å, 1.7 μm, 2.1 × 50 mm) coupled to a Xevo TQ-XS triple-stage quadrupole mass spectrometer equipped with a Z-spray heated ESI source (Waters, Milford, MA, USA). Injection volumes were 20 μL. Chromatography was performed with mixtures of aqueous eluent A consisting of water–acetonitrile–formic acid, 95:5:0.1% *v*/*v*, and organic eluent B consisting of acetonitrile with 0.1% (*v*/*v*) formic acid. Elution and separation were achieved with a linear gradient from 100% A to 95% A/5% B within 2 min, followed by a linear gradient within 3 min to 70% A/30% B. The flow rate was 0.5 mL/min. Transitions used for monitoring multiple reactions were *m*/*z* 885.65 **→**
*m*/*z* 665.50 for LTI, *m*/*z* 890.2 **→**
*m*/*z* 669.08 for LTI-IS, *m*/*z* 522.9 **→**
*m*/*z* 644.39 for SSL and *m*/*z* 526.54 **→**
*m*/*z* 644.53 for SSL-IS. The dwell time for each transition was 25 ms. Mass spectrometric parameters in the positive ion mode were a capillary voltage of 500 V and a cone voltage of 20 V. Collision energy was 26 eV for LTI and LTI-IS and 20 eV for SSL and SSL-IS. Calibration curves were fitted by linear regression with a 1/x2 weighting and the calculation of concentrations was performed using the TargetLynx software (v 4.2; Waters, Milford, MA, USA). The determined protein concentrations were normalized to the total amount of protein in the sample.

### 4.10. UPLC-MS/MS Method Validation

Validation of the assay was performed in three analytical runs to assess inter- and intraday parameters via a determination of their accuracy and precision, linearity, recovery, and matrix effect using the applicable limits of the ICH M10 [73]. Analytical runs for validation purposes included six-fold determinations of concentrations at LLOQ, low QC, mid QC, and high QC. Accuracy was evaluated as a percent of the nominal value. The corresponding precision was determined as the respective coefficient of variation. Recovery of the solid phase extraction was assessed at low, mid, and high QC concentrations using the peak areas of processed QCs, compared to blank matrix samples spiked after extraction. The matrix effect was determined from blank matrix samples spiked after processing with pure solvent solutions of a similar volume containing the respective amounts of peptides [74]. In addition, for SSL, the precision and accuracy at a concentration of 12.5 pg/mL were determined using two independent batches with six replicates.

### 4.11. Transporter Activity

ABCC2 activity was determined in HepG2 cells by measuring the efflux of dichlorofluorescein (DCF) [75,76]. Cells were plated in 6-well plates (1.6 × 10^6^ cells/well), cultured, and treated with rifampicin and hypericin, as described in Section 4.2. Once the treatment was finalized, cells were incubated with 5(6)-carboxy-2′,7′-dichlorofluorescein diacetate (10 µM) in the presence or absence of MK571 (20 µM, ABCC inhibitor, [75]) for 30 min. Once inside the cell, 5(6)-carboxy-2′,7′-dichlorofluorescein diacetate is hydrolyzed by intracellular esterases to DCF, which is transported by ABCC2 [75,76]. After the incubation, the amount of DCF transported to the extracellular medium was determined by spectrofluorometry (λ_excitation_: 485 nm; λ_emission_: 525 nm). ABCC2-specific DCF efflux was calculated as the difference between the total amount of DCF that was transported (determined in the absence of MK571) and the ABCC2-independent DCF efflux (determined in the presence of MK571).

ABCG2 activity was determined in MCF7 cells by measuring the efflux of DOX, as previously described [77]. Cells were plated in 6-well plates (5.7 × 10^5^ cells/well) and treated with rifampicin, as described in Section 4.2. Afterwards, the treatment medium was removed and cells were incubated with DOX (10 µM, 1 h). After loading the cells with the substrate, cells were incubated for 1 h in HBSS at 37 °C in the presence or absence of FTC (10 µM), as an ABCG2 inhibitor [77]. The amount of DOX transported to the extracellular medium was determined by spectrofluorometry (λ_excitation_: 485 nm; λ_emission_: 538 nm). ABCG2-specific DOX efflux was calculated as the difference between the total amount of DOX that was transported (determined in the absence of FTC) and the ABCG2-independent DOX efflux (determined in the presence of FTC).

### 4.12. Viability Assay

Cell viability was determined for treatments resulting in transporter regulation in both cells and EVs using the crystal violet assay [77]. For this purpose, HepG2 (5.6 × 10^4^ cells/well) and MCF7 cells (1.8 × 10^4^ cells/well) were plated in 96-well plates and treated, as described in Section 4.2. Afterwards, viable cells were stained with crystal violet, and absorbance (555 nm) was determined by spectrophotometry.

### 4.13. Statistical Analysis

Data (mean ± S.D.) were analyzed with GraphPad Prism (v 9.0, GraphPad Software, La Jolla, CA, USA). Statistical comparisons were performed by applying the One-Way ANOVA test followed by Tukey’s post-hoc test for more than two experimental groups. Transporter expression in plasma EVs before and at the end of the treatment was compared using a paired Student’s *t*-test. Significance was set at *p* < 0.05.

## 5. Conclusions

In the present work, we demonstrated the dynamic association between the protein expression levels of ABCC2 in HepG2 cells and their derived EVs after exposure to two regulatory compounds. ABCG2 levels in MCF7 EVs did not reflect the expression in the cells. Furthermore, we demonstrated the presence of ABCC2 and ABCG2 in plasma EVs from two different cohorts of healthy volunteers. Our findings suggest the potential of EVs to estimate the expression levels of at least ABCC2 in HepG2 cells during the course of a treatment. Further studies in more complex in vitro models (e.g., primary human hepatocytes, organoids) and in larger cohorts are now required to confirm the relevance of these findings for clinical practice.

## Figures and Tables

**Figure 1 ijms-25-04118-f001:**
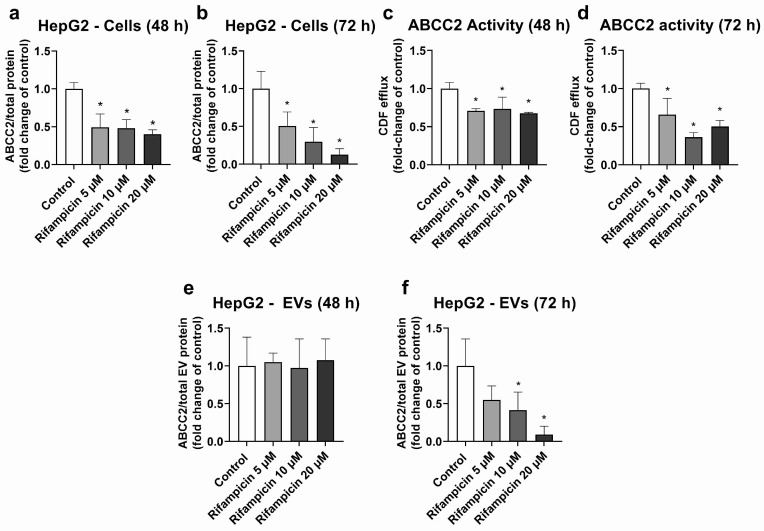
Expression and activity of ABCC2 in HepG2 cells treated with rifampicin (5–20 µM) and their derived EVs. ABCC2 expression was quantified by UPLC-MS/MS in HepG2 cell lysates after treatment with rifampicin for 48 (**a**) and 72 h (**b**). ABCC2 activity was determined in HepG2 cells treated with rifampicin for 48 (**c**) and 72 h (**d**) by measuring the efflux of dichlorofluorescein (DCF) by spectrofluorometry. ABCC2 expression was determined by UPLC-MS/MS in EVs from cells treated with rifampicin for 48 (**e**) and 72 h (**f**). Results (mean ± S.D.) are expressed as fold change in the expression or activity in the control (untreated cells). *: different from control, *p* < 0.05, *n* ≥ 3.

**Figure 2 ijms-25-04118-f002:**
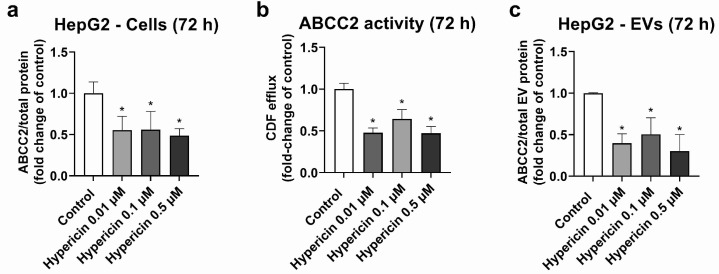
Expression and activity of ABCC2 in HepG2 cells treated with hypericin (0.01–0.5 µM, 72 h) and their derived EVs. ABCC2 expression in HepG2 cell lysates (**a**) was quantified by UPLC-MS/MS and ABCC2 activity (**b**) by spectrofluorometry. ABCC2 expression was determined in EVs from cells treated with rifampicin for 72 h by UPLC-MS/MS (**c**). Results (mean ± S.D.) are expressed as fold change in the expression or activity in the control (untreated cells). *: different from control, *p* < 0.05, *n* ≥ 3.

**Figure 3 ijms-25-04118-f003:**
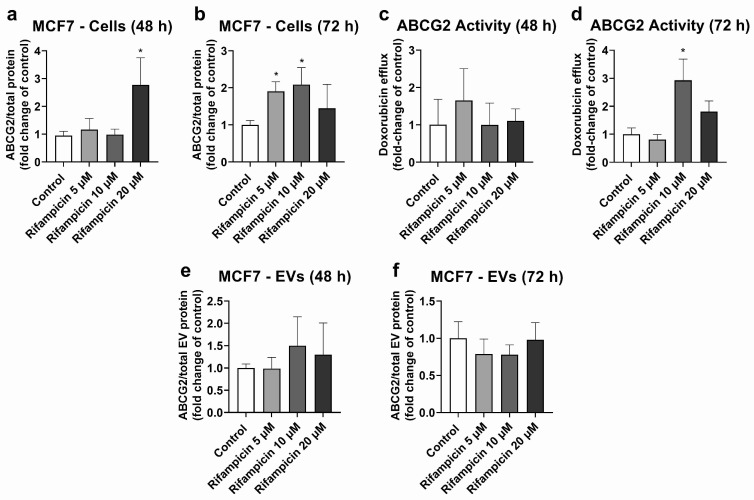
Expression and activity of ABCG2 in MCF7 cells treated with rifampicin (5–20 µM) and their derived EVs. ABCG2 expression was quantified in MCF7 cell lysates after treatment with rifampicin for 48 (**a**) and 72 h (**b**) by UPLC-MS/MS. ABCG2 activity was determined in MCF7 cells treated with rifampicin for 48 (**c**) and 72 h (**d**) by measuring the efflux of doxorubicin by spectrofluorometry. ABCG2 expression was determined by UPLC-MS/MS in EVs from MCF7 cells treated with rifampicin for 48 (**e**) and 72 h (**f**). Results (mean ± S.D.) are expressed as fold change in the expression or activity in the control (untreated cells). *: different from control, *p* < 0.05, *n* = 3–6.

**Figure 4 ijms-25-04118-f004:**
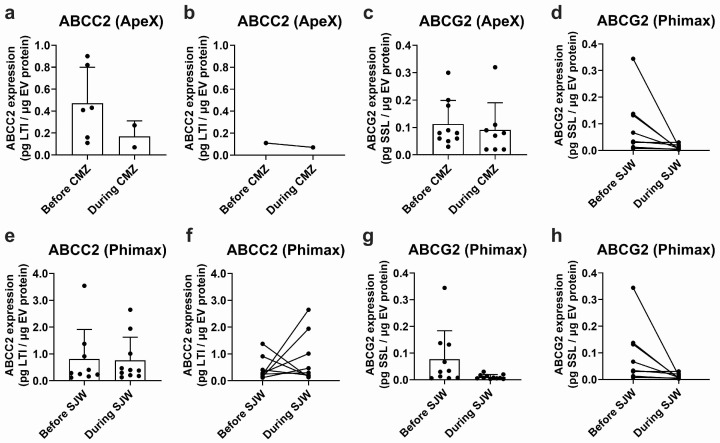
Expression of ABCC2 and ABCG2 in plasma EVs. ABCC2 (**a**) and ABCG2 (**c**) expression in plasma EVs from the ApeX study before and at the end of the treatment with carbamazepine (i.e., during CMZ, 200 mg/day, 21 days, p.o) and ABCC2 (**e**) and ABCG2 (**g**) expression in plasma EVs from the Phimax study before and at the end of the treatment with St. John’s wort (i.e., during SJW, 3 × 300 mg/day, 13 days, p.o.) were determined by UPLC-MS/MS. Paired ABCC2 (**b**) and ABCG2 (**d**) expression data are presented for each participant of the ApeX study with detectable transporter expression before and at the end of the treatment with CMZ. Paired ABCC2 (**f**) and ABCG2 (**h**) expression data are presented for each participant of the Phimax study with detectable transporter expression before and at the end of the treatment with SJW. Data (mean ± S.D.) are presented as pg. of the surrogate peptide. *n* = 12 (ApeX) and *n* = 13 (Phimax).

**Table 1 ijms-25-04118-t001:** Summary of quality control results for the ABCC2 surrogate peptide (LTI) and the ABCG2 surrogate peptide (SSL).

	LTI (ABCC2 Surrogate Peptide)				SSL (ABCG2 Surrogate Peptide)			
	25 pg/mL	75 pg/mL	37.5 ng/mL	75 ng/mL	25 pg/mL	75 pg/mL	37.5 ng/mL	75 ng/mL
Within Batch								
Batch 1								
Mean (pg/mL)	23.3	65.4	36,277	70,934	25.1	67.3	34,166	66,712
Accuracy (%)	93.3	87.2	96.7	94.6	99.6	89.8	91.1	89.0
Precision (% CV)	18.8	2.67	2.81	1.92	4.01	4.61	2.74	2.22
Batch 2								
Mean (pg/mL)	22.8	67.7	34,310	68,602	24.2	72.2	33,562	67,028
Accuracy (%)	91.2	90.3	91.5	91.5	96.9	96.3	89.5	89.4
Precision (% CV)	19.7	4.57	3.49	2.12	7.89	2.32	2.74	1.89
Batch 3								
Mean (pg/mL)	24.5	71.4	34,408	67,972	23.0	72.6	32,979	65,673
Accuracy (%)	97.8	95.2	91.8	90.6	92.0	96.7	88.0	87.6
Precision (% CV)	15.2	5.11	2.58	2.03	9.21	4.85	1.41	1.10
Batch to batch								
Mean (pg/mL)	23.6	68.2	34,998	69,169	24.0	71.1	33,569	66,471
Accuracy (%)	94.4	90.9	93.3	92.2	96.2	94.7	89.5	88.6
Precision (% CV)	16.4	5.45	3.86	2.68	7.81	5.02	2.69	1.92

Three independent validation batches (six replicates each) were used. CV: coefficient of variation.

## Data Availability

All data are presented in the manuscript or as Appendix A.

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
