# Peer review of "Extracellular Vesicles as Surrogates for the Regulation of the Drug Transporters ABCC2 (MRP2) and ABCG2 (BCRP)"

_ijms, 2024, doi:10.3390/ijms25074118_

Round 1
Reviewer 1 Report
Comments and Suggestions for Authors
In this manuscript, the authors developed and validated a UPLC-MS/MS method to quantify ABCC2 and ABCG2 in extracellular vericles (EVs) from cell culture and plasma. What’s more, an association between ABCC2 protein levels and transporter activity in HepG2 cells treated with rifampicin and hypericin and their derived EVs was observed. However, so many key points were missing in this manuscript. Therefore, I recommend that these essential experiments must be conducted and the results should be integrated into the manuscript before considering its acceptance for publication.
Major issues should be addressed.
1. In the introduction of the manuscript, the description of ABCG2 is much too redundant, and the authors are requested to make appropriate reductions in this section.
2. Whether the data presented in Figure 1 can be replaced with a visualized statistical graph, and the authors are advised to change this.
3. In the EV characterization section of the manuscript, in addition to the study of markers, it is likewise necessary to characterize the morphology of EVs.
4. In the manuscript, please keep the layout of Figures 1 and 3 consistent with the other data plots.
5. In Table S2 of the manuscript, the authors were asked to provide a brief description of how the concentration of ABCC2 was determined.
6. In Figure 4b of the manuscript, the authors are asked to explain why there is only one sample data in the study about ABCC2 expression.
7. In the manuscript, the authors did not investigate the half-life of EV, please perform additional experiments in this section.
8. In the abstract section of the manuscript, the authors mention that both ABCC2 and ABCG2 can determine the pharmacokinetics of many drugs, but the paper does not cover the relevant evaluation experiments.
9. Pay attention to the correct use of punctuation.
Comments on the Quality of English LanguageThe manuscript should be carefully checked and revised to avoid the spelling, expression and grammar errors.
Author Response
Reviewer 1
In this manuscript, the authors developed and validated a UPLC-MS/MS method to quantify ABCC2 and ABCG2 in extracellular vericles (EVs) from cell culture and plasma. What’s more, an association between ABCC2 protein levels and transporter activity in HepG2 cells treated with rifampicin and hypericin and their derived EVs was observed. However, so many key points were missing in this manuscript. Therefore, I recommend that these essential experiments must be conducted and the results should be integrated into the manuscript before considering its acceptance for publication.
Major issues should be addressed.
1.In the introduction of the manuscript, the description of ABCG2 is much too redundant, and the authors are requested to make appropriate reductions in this section.
Answer: According to the Reviewer’s suggestion, we have shortened the description of ABCG2.
2. Whether the data presented in Figure 1 can be replaced with a visualized statistical graph, and the authors are advised to change this.
Answer: Here, we are not sure whether the Reviewer refers to Figure 1 or Supplementary Figure S1. Figure 1 is already provided with statistics. Figure S1 (western blot for EV markers) aims to show the presence or absence of the EV markers CD63 and CD81. No quantitative information shall be obtained from this figure.
3. In the EV characterization section of the manuscript, in addition to the study of markers, it is likewise necessary to characterize the morphology of EVs.
Answer: in addition to the study of markers (Fig. S1), we performed dynamic light scattering measurements to confirm the size of EVs for all our samples (Table S1 for cell culture EVs and lines 143-145 for plasma EVs). Considering that our EV samples were purified by a combination of methods delivering high purity samples (ultrafiltration + size exclusion chromatography), we believe that western blot and dynamic light scattering data should be enough to characterize our samples. Besides, the form of EVs isolated eluted from qEV size exclusion chromatography columns, as used in our study, was already demonstrated by other authors (Chiaradia et al., Eur J Cell Biol, 102: 151285, 2023).
4. In the manuscript, please keep the layout of Figures 1 and 3 consistent with the other data plots.
Answer: We regret, but we do not know what the Reviewer refers to.
5. In Table S2 of the manuscript, the authors were asked to provide a brief description of how the concentration of ABCC2 was determined.
Answer: The ABBC2 expression is expressed as fold-change of the control and was determined as described in the section 4.9 of the materials and methods. In response to this comment, a mention to this section has been added in the legend of table S2 to clarify this.
6. In Figure 4b of the manuscript, the authors are asked to explain why there is only one sample data in the study about ABCC2 expression.
Answer: Figure 4b shows the change in ABCC2 expression for each participant due to the treatment. Only one study participant showed detectable ABCC2 expression in plasma EVs both before and at the end of the treatment. The individual ABCC2 expression data before and after treatment is presented in Figure 4a.
7. In the manuscript, the authors did not investigate the half-life of EV, please perform additional experiments in this section.
Answer: EV stability has been already analyzed by another group (reference 51, lines 282) and is thus not necessary to be repeated.
8. In the abstract section of the manuscript, the authors mention that both ABCC2 and ABCG2 can determine the pharmacokinetics of many drugs, but the paper does not cover the relevant evaluation experiments.
Answer: We are not sure, whether we understood the Reviewer correctly, but the statement in the abstract refers to the well-known role of these transporters in drug pharmacokinetics as described in the introduction.
9. Pay attention to the correct use of punctuation.
Answer: In response to this comment, the manuscript has been revised to adapt the use of punctuation.
Reviewer 2 Report
Comments and Suggestions for Authors
The manuscript entitled “Extracellular vesicles as surrogates for the regulation of the 2 drug transporters ABCC2 (MRP2) and ABCG2 (BCRP)” by Juan Pablo Rigalli et al. developed a UPLC-MS/MS method to measure ABCC and ABCG2 protein level in extracellular vesicles as a surrogate for human drug transporter analysis. However, the experimental result only shows a correlation in HepG2 cell lines, which limit the potential application of this method. Key experiments are lacking to validate the transporter protein level. Here are the major concerns.
1. As author mentioned, extracellular vesicles have various origins from different organs. The EV generation rate, amount as well as the EV content are significantly different of different origins. In this study the correlation was only seen in HepG2 cells, which is a liver cancer cell line and cannot be directly translated to the human hepatocytes. To demonstrate the potential of this method, mouse or human hepatocyte should be used.
2. Author only tested HepG2 and MCF7 cells, and only HepG2 cells show good correlation. The result cannot demonstrate whether the method can be applied to different cell lines and more complicated organs and tissues.
3. The UPLC-MS/MS method have achieved with high accuracy and sensitivity, however; the transporter protein as well as EVs could be lost and degraded during the purification and concentration process, to validate this method on cell and EVs, another biochemical method like ELISA or quantitative WB need to be used. Without such validation, the result cannot be verified.
4. The biological experiments in the manuscript were not well designed and misleading. For example, ALL the western blot images are dirty and hard to quantify. More importantly, if the WB image is derived by chemiluminescence, why the ladder also has band in the image as the ladder doesn’t bind IgG HRP antibody. Also, the transporter assay need to include a positive control as the result from different groups have huge variation.
Author Response
Reviewer 2
The manuscript entitled “Extracellular vesicles as surrogates for the regulation of the 2 drug transporters ABCC2 (MRP2) and ABCG2 (BCRP)” by Juan Pablo Rigalli et al. developed a UPLC-MS/MS method to measure ABCC and ABCG2 protein level in extracellular vesicles as a surrogate for human drug transporter analysis. However, the experimental result only shows a correlation in HepG2 cell lines, which limit the potential application of this method. Key experiments are lacking to validate the transporter protein level. Here are the major concerns.
1. As author mentioned, extracellular vesicles have various origins from different organs. The EV generation rate, amount as well as the EV content are significantly different of different origins. In this study the correlation was only seen in HepG2 cells, which is a liver cancer cell line and cannot be directly translated to the human hepatocytes. To demonstrate the potential of this method, mouse or human hepatocyte should be used.
Answer: The aim of this work was to deliver a first proof of concept, whether protein levels in cells and in EVs may change in the same direction upon exposure to drugs. Performing the experiments described in our work required large amounts of cells in order to collect enough EV material with enough purity for UPLC-MS/MS analysis. The costs of using human hepatocytes would have been extremely high to perform a first proof of concept. This should be performed in follow up studies. Mouse hepatocytes do not constitute the most suitable model since transporter regulation by drugs is frequently species-specific. In response to this comment, we have highlighted primary human hepatocytes as the model for the next studies in the conclusions section (see line 296).
2. Author only tested HepG2 and MCF7 cells, and only HepG2 cells show good correlation. The result cannot demonstrate whether the method can be applied to different cell lines and more complicated organs and tissues.
Answer: in line with our answer to the previous comment, this study represented a first proof-of-concept. This was the first time that the correlation between ABC transporters in cells and EVs at the protein level was investigated. For this purpose, we chose simpler and easily accessible models. To disclose this limitation, we added a statement in the discussion (see lines xxx): “Although transporter half-life and EV stability should be strictly taken into consideration to detect transporter down-regulation during the course of a treatment, our findings clearly highlight the potential of EVs as surrogates for ABCC2 transporter levels in HepG2 cells. Studies in more complex models (e.g., primary hepatocytes) should be performed to elucidate the translational potential of these findings”.
3. The UPLC-MS/MS method have achieved with high accuracy and sensitivity, however; the transporter protein as well as EVs could be lost and degraded during the purification and concentration process, to validate this method on cell and EVs, another biochemical method like ELISA or quantitative WB need to be used. Without such validation, the result cannot be verified.
Answer: Thank you for this comment. Degradation of the sample during the purification and concentration process cannot be totally ruled out. Analyzing the samples by other methods such as ELISA or quantitative WB, as suggested by the Reviewer, would also require EV and protein isolation and concentration. Therefore, if occurring, sample lose or degradation would also affect the results obtained by using other biochemical methods.
4. The biological experiments in the manuscript were not well designed and misleading. For example, ALL the western blot images are dirty and hard to quantify. More importantly, if the WB image is derived by chemiluminescence, why the ladder also has band in the image as the ladder doesn’t bind IgG HRP antibody. Also, the transporter assay need to include a positive control as the result from different groups have huge variation.
Answer: All the western blots were aimed at obtaining qualitative information. In the supplementary material S1, the presence of EV markers CD63 and CD81 was investigated. This constitutes a proof of the presence of EVs in the elution fraction of the size exclusion chromatography. No quantitative information shall be obtained from these data.
In Figure S8, we added western blot data to confirm the UPLC-MS/MS results showing higher ABCC2 expression in HepG2 cells (Fig S3 and S4), which justify the study of ABCC2 in HepG2 cells (and not in MCF7 cells, where the transporter is under the limit of detection). Samples from other (unrelated) cell lines were loaded in the same western blot. We apologize if this led to confusion but it is not allowed for us to crop bands to show only MCF7 and HepG2 samples.
The signal from the ladder is likely attributed to the unspecific interaction between the secondary antibody and the components of the ladder. This is a very common and useful phenomenon in western blot and does not affect the conclusions obtained from these experiments.
Round 2
Reviewer 2 Report
Comments and Suggestions for Authors
I appreciate author's reply to the comments. The UPLC-MS/MS method developed in this manuscript could be good fit for a bioanalytical method publication, however; the application of this described method, using extracellular vesicles (EV) as surrogate for transporter analysis, need to be further evaluated.
Certainly harvesting and purifying enough EVs from cell culture is expensive and time consuming, however; it is actually one of the major limitation of using EV in the pre-clinical and clinical diagnosis. Consider the variability of EV origin and the complexity in vivo, whether the Extracellular vesicles as surrogates could represent the regulation of the drug transporters ABCC2 (MRP2) and ABCG2 remains questionable.
Author Response
I appreciate author's reply to the comments. The UPLC-MS/MS method developed in this manuscript could be good fit for a bioanalytical method publication, however; the application of this described method, using extracellular vesicles (EV) as surrogate for transporter analysis, need to be further evaluated.
Certainly harvesting and purifying enough EVs from cell culture is expensive and time consuming, however; it is actually one of the major limitation of using EV in the pre-clinical and clinical diagnosis. Consider the variability of EV origin and the complexity in vivo, whether the Extracellular vesicles as surrogates could represent the regulation of the drug transporters ABCC2 (MRP2) and ABCG2 remains questionable.
Answer: We agree with the Reviewer on the need for further experiments in different models in vitro, in vivo and under different conditions to confirm the use of EVs as surrogate for ABC transporter expression and activity in clinical diagnosis.
The use of EVs as source of biomarkers, in general, requires extensive testing in different models and cohorts and a confirmation is rarely achieved within one study. In this regard, our manuscript delivers the first piece of evidence showing the association, or the lack of association in the case of ABCG2, between transporter protein levels in cells and EVs. We consider our work only as a starting point and are fully aware that this evidence is not enough to translate them into the clinical practice. This is clearly stated in the discussion section in lines 295-297, 317-318, 351-354 (added in this revision round) and in the conclusions (lines 566-568). We have also modified the last sentence of the abstract and lines 566-568 to specify that associations were observed in HepG2 cells and to avoid more general claims.
Round 3
Reviewer 2 Report
Comments and Suggestions for Authors
Thanks for the revision. The manuscript could be consider for publication.